# A Direct $\tilde{O}(1/\epsilon)$ Iteration Parallel Algorithm for Optimal Transport

**Arun Jambulapati, Aaron Sidford, and Kevin Tian**
Stanford University
{jmblpati,sidford,kjtian}@stanford.edu

## Abstract

Optimal transportation, or computing the Wasserstein or "earth mover's" distance between two $n$-dimensional distributions, is a fundamental primitive which arises in many learning and statistical settings. We give an algorithm which solves the problem to additive $\epsilon$ accuracy with $\tilde{O}(1/\epsilon)$ parallel depth and $\tilde{O}\left(n^2/\epsilon\right)$ work. [BJKS18, Qua19] obtained this runtime through reductions to positive linear programming and matrix scaling. However, these reduction-based algorithms use subroutines which may be impractical due to requiring solvers for second-order iterations (matrix scaling) or non-parallelizability (positive LP). Our methods match the previous-best work bounds by [BJKS18, Qua19] while either improving parallelization or removing the need for linear system solves, and improve upon the previous best first-order methods running in time $\tilde{O}(\min(n^2/\epsilon^2, n^{2.5}/\epsilon))$ [DGK18, LHJ19]. We obtain our results by a primal-dual extragradient method, motivated by recent theoretical improvements to maximum flow [She17].

## 1   Introduction

Optimal transport is playing an increasingly important role as a subroutine in tasks arising in machine learning [ACB17], computer vision [BvdPPH11, SdGP+15], robust optimization [EK18, BK17], and statistics [PZ16]. Given these applications for large scale learning, designing algorithms for efficiently approximately solving the problem has been the subject of extensive recent research [Cut13, AWR17, GCPB16, CK18, DGK18, LHJ19, BJKS18, Qua19].

Given two vectors $r$ and $c$ in the $n$-dimensional probability simplex $\Delta^n$ and a cost matrix $C \in \mathbb{R}_{\geq 0}^{n \times n}$[1], the optimal transportation problem is

$$\min_{X \in \mathcal{U}_{r,c}} \langle C, X \rangle, \quad \text{where} \quad \mathcal{U}_{r,c} \stackrel{\text{def}}{=} \left\{ X \in \mathbb{R}_{\geq 0}^{n \times n}, \ X\mathbf{1} = r, \ X^\top \mathbf{1} = c \right\}. \tag{1}$$

This problem arises from defining the *Wasserstein* or *Earth mover's* distance between discrete probability measures $r$ and $c$, as the cheapest coupling between the distributions, where the cost of the coupling $X \in \mathcal{U}_{r,c}$ is $\langle C, X \rangle$. If $r$ and $c$ are viewed as distributions of masses placed on $n$ points in some space (typically metric), the Wasserstein distance is the cheapest way to move mass to transform $r$ into $c$. In (1), $X$ represents the transport plan ($X_{ij}$ is the amount moved from $r_i$ to $c_j$) and $C$ represents the cost of movement ($C_{ij}$ is the cost of moving mass from $r_i$ to $c_j$).

Throughout, the value of (1) is denoted OPT. We call $\hat{X} \in \mathcal{U}_{r,c}$ an *$\epsilon$-approximate transportation plan* if $\langle C, \hat{X} \rangle \leq \text{OPT} + \epsilon$. Our goal is to design an efficient algorithm to produce such a $\hat{X}$.

## 1.1 Our Contributions

Our main contribution is an algorithm running in $\tilde{O}(\|C\|_{\max}/\epsilon)$ parallelizable iterations[2] and $\tilde{O}(n^2\|C\|_{\max}/\epsilon)$ total work producing an $\epsilon$-approximate transport plan.

Matching runtimes were given in the recent work of [BJKS18, Qua19]. Their runtimes were obtained via reductions to matrix scaling and positive linear programming, each well-studied problems in theoretical computer science. However, the matrix scaling algorithm is a second-order Newton-type method which makes calls to structured linear system solvers, and the positive LP algorithm is not parallelizable (i.e. has depth polynomial in dimension). These features potentially limit the practicality of these algorithms. The key remaining open question this paper addresses is, *is there an efficient first-order, parallelizable algorithm for approximating optimal transport?* We answer this affirmatively and give an efficient, parallelizable primal-dual first-order method; the only additional overhead is a scheme for implementing steps, incurring roughly an additional $\log \epsilon^{-1}$ factor.

Our approach heavily leverages the recent improvement to the maximum flow problem, and more broadly two-player games on a simplex ($\ell_1$ ball) and a box ($\ell_\infty$ ball), due to the breakthrough work of [She17]. First, we recast (1) as a minimax game between a box and a simplex, proving correctness via a rounding procedure known in the optimal transport literature. Second, we show how to adapt the dual extrapolation scheme under the weaker convergence requirements of area-convexity, following [She17], to obtain an approximate minimizer to our primal-dual objective in the stated runtime. En route, we slightly simplify analysis in [She17] and relate it more closely to the existing extragradient literature.

Finally, we give preliminary experimental evidence showing our algorithm can be practical, and highlight some open directions in bridging the gap between theory and practice of our method, as well as accelerated gradient schemes [DGK18, LHJ19] and Sinkhorn iteration.

## 1.2 Previous Work

**Optimal Transport.** The problem of giving efficient algorithms to find $\epsilon$-approximate transport plans $\hat{X}$ which run in nearly linear time[3] has been addressed by a line of recent work, starting with [Cut13] and improved upon in [GCPB16, AWR17, DGK18, LHJ19, BJKS18, Qua19]. We briefly discuss their approaches here.

Works by [Cut13, AWR17] studied the Sinkhorn algorithm, an alternating minimization scheme. Regularizing (1) with an $\eta^{-1}$ multiple of entropy and computing the dual, we arrive at the problem

$$\min_{x,y\in\mathbb{R}^n} \mathbf{1}^\top B_{\eta C}(x,y)\mathbf{1} - r^\top x - c^\top y \quad \text{where} \quad B_{\eta C}(x,y)_{ij} = e^{x_i+y_j-\eta C_{ij}}.$$

This problem is equivalent to computing diagonal scalings $X$ and $Y$ for $M = \exp(-\eta C)$ such that $XMY$ has row sums $r$ and column sums $c$. The Sinkhorn iteration alternates fixing the row sums and the column sums by left and right scaling by diagonal matrices until an approximation of such scalings is found, or equivalently until $XMY$ is close to being in $\mathcal{U}_{r,c}$.

As shown in [AWR17], we can round the resulting almost-transportation plan to a transportation plan which lies in $\mathcal{U}_{r,c}$ in linear time, losing at most $2\|C\|_{\max}(\|X\mathbf{1} - r\|_1 + \|X^\top\mathbf{1} - c\|_1)$ in the objective. Further, [AWR17] showed that $\tilde{O}(\|C\|_{\max}^3/\epsilon^3)$ iterations of this scheme sufficed to obtain a matrix which $\epsilon/\|C\|_{\max}$-approximately meets the demands in $\ell_1$ with good objective value, by analyzing it as an instance of mirror descent with an entropic regularizer. The same work proposed an alternative algorithm, Greenkhorn, based on greedy coordinate descent. [DGK18, LHJ19] showed that $\tilde{O}\left(\|C\|_{\max}^2/\epsilon^2\right)$ iterations, corresponding to $\tilde{O}\left(n^2\|C\|_{\max}^2/\epsilon^2\right)$ work, suffice for both Sinkhorn and Greenkhorn, the current state-of-the-art for this line of analysis.

An alternative approach based on first-order methods was studied by [DGK18, LHJ19]. These works considered minimizing an entropy-regularized Equation 1; the resulting weighted softmax function is prevalent in the literature on approximate linear programming [Nes05], and has found similar

applications in near-linear algorithms for maximum flow [She13, KLOS14, ST18] and positive linear programming [You01, AO15]. An unaccelerated algorithm, viewable as $\ell_\infty$ gradient descent, was analyzed in [DGK18] and ran in $\tilde{O}(\|C\|_{\max}/\epsilon^2)$ iterations. Further, an accelerated algorithm was discussed, for which the authors claimed an $\tilde{O}(n^{1/4}\|C\|_{\max}^{0.5}/\epsilon)$ iteration count. [LHJ19] showed that the algorithm had an additional dependence on a parameter as bad as $n^{1/4}$, roughly due to a gap between the $\ell_2$ and $\ell_\infty$ norms. Thus, the state of the art runtime in this line is the better of $\tilde{O}\left(n^{2.5}\|C\|_{\max}^{0.5}/\epsilon\right)$, $\tilde{O}\left(n^2\|C\|_{\max}/\epsilon^2\right)$ operations. The dependence on dimension of the former of these runtimes matches that of the linear programming solver of [LS14, LS15], which obtain a polylogarithmic dependence on $\epsilon^{-1}$, rather than a polynomial dependence; thus, the question of obtaining an accelerated $\epsilon^{-1}$ dependence without worse dimension dependence remained open.

This was partially settled in [BJKS18, Qua19], which studied the relationship of optimal transport to fundamental algorithmic problems in theoretical computer science, namely positive linear programming and matrix scaling, for which significantly-improved runtimes have been recently obtained [AO15, ZLdOW17, CMTV17]. In particular, they showed that optimal transport could be reduced to instances of either of these objectives, for which $\tilde{O}\left(\|C\|_{\max}/\epsilon\right)$ iterations, each of which required linear $O(n^2)$ work, sufficed. However, both of these reductions are based on black-box methods for which practical implementations are not known; furthermore, in the case of positive linear programming a parallel $\tilde{O}(1/\epsilon)$-iteration algorithm is not known. [BJKS18] also showed any polynomial improvement to the runtime of our paper in the dependence on either $\epsilon$ or $n$ would result in maximum-cardinality bipartite matching in dense graphs faster than $\tilde{O}(n^{2.5})$ without fast matrix multiplication [San09], a fundamental open problem unresolved for almost 50 years [HK73].

| Year | Author | Complexity | Approach | 1st-order | Parallel |
|---|---|---|---|---|---|
| 2015 | [LS15] | $\tilde{O}(n^{2.5})$ | Interior point | No | No |
| 2017-19 | [AWR17] | $\tilde{O}(n^2\|C\|_{\max}^2/\epsilon^2)$ | Sink/Greenkhorn | Yes | Yes |
| 2018 | [DGK18] | $\tilde{O}(n^2\|C\|_{\max}^2/\epsilon^2)$ | Gradient descent | Yes | Yes |
| 2018-19 | [LHJ19] | $\tilde{O}(n^{2.5}\|C\|_{\max}/\epsilon)$ | Acceleration | Yes | Yes |
| 2018 | [BJKS18] | $\tilde{O}(n^2\|C\|_{\max}/\epsilon)$ | Matrix scaling | No | Yes |
| 2018-19 | [BJKS18, Qua19] | $\tilde{O}(n^2\|C\|_{\max}/\epsilon)$ | Positive LP | Yes | No |
| 2019 | This work | $\tilde{O}(n^2\|C\|_{\max}/\epsilon)$ | Dual extrapolation | Yes | Yes |

Table 1: Optimal transport algorithms. Algorithms using second-order information use potentially-expensive SDD system solvers; the runtime analysis of Sink/Greenkhorn is due to [DGK18, LHJ19].

Specializations of the transportation problem to $\ell_p$ metric spaces or arising from geometric settings have been studied [SA12, AS14, ANOY14]. These specialized approaches seem fundamentally different than those concerning the more general transportation problem.

Finally, we note recent work [ABRW18] showed the promise of using the Nyström method for low-rank approximations to achieve speedup in theory and practice for transport problems arising from specific metrics. As our method is based on matrix-vector operations, where low-rank approximations may be applicable, we find it interesting to see if our method can be combined with these improvements.

*Remark.* During the revision process for this work, an independent result [LMR19] was published to arXiv, obtaining improved runtimes for optimal transport via a combinatorial algorithm. The work obtains a runtime of $\tilde{O}(n^2\|C\|_{\max}/\epsilon + n\|C\|_{\max}^2/\epsilon^2)$, which is worse than our runtime by a low-order term. Furthermore, it does not appear to be parallelizable.

**Box-simplex objectives.** Our main result follows from improved algorithms for bilinear minimax problems over one simplex domain and one box domain developed in [She17]. This fundamental minimax problem captures $\ell_1$ and $\ell_\infty$ regression over a simplex and box respectively, and inspired the development of conjugate smoothing [Nes05] as well as mirror prox / dual extrapolation [Nem04, Nes07]. These latter two approaches are extragradient methods (using two gradient operations per iteration rather than one) for approximately solving a family of problems, which includes convex minimization and finding a saddle point to a convex-concave function. These methods simulate backwards Euler discretization of the gradient flow, similar to how mirror descent simulates forwards

Euler discretization [DO19]. The role of the extragradient step is a fixed point iteration (of two steps) which is a good approximation of the backwards Euler step when the operator is Lipschitz.

Nonetheless, the analysis of [Nem04, Nes07] fell short in obtaining a $1/T$ rate of convergence without worse dependence on dimension for these domains, where $T$ is the iteration count (which would correspond to a $\tilde{O}(1/\epsilon)$ runtime for approximate minimization). The fundamental barrier was that over a box, any strongly-convex regularizer in the $\ell_\infty$ norm has a dimension-dependent domain size (shown in [ST18]). This barrier can also be viewed as the reason for the worse dimension dependence in the accelerated scheme of [DGK18, LHJ19].

The primary insight of [She17] was that previous approaches attempted to regularize the schemes of [Nem04, Nes07] with separable regularizers, i.e. the sum of a regularizer which depends only on the primal block and one which depends only on the dual. If, say, the domain of the primal block was a box, then such a regularization scheme would run into the $\ell_\infty$ barrier and incur a worse dependence on dimension. However, by more carefully analyzing the requirements of these algorithms, [She17] constructed a non-separable regularizer with small domain size, satisfying a property termed *area-convexity* which sufficed for provable convergence of dual extrapolation [Nes07]. Interestingly, the property seems specialized to dual extrapolation and not mirror prox [Nem04].

## 2   Overview

First, in Section 2.1 we first describe a reformulation of (1) as a primal-dual objective, which we solve approximately in Section 3. Then in Section 2.2 we give additional notation critical for our analysis[4]. In Section 3 we leverage this to give an overview of our main algorithm.

### 2.1   $\ell_1$-regression formulation

We adapt the view of [BJKS18, Qua19] of the objective (1) as a positive linear program. Let $d$ be the (vectorized) cost matrix $C$ associated with the instance and let $\Delta^{n^2}$ be the $n^2$ dimensional simplex[5]. We recall $r, c$ are specified row and column sums with $\mathbf{1}^\top r = \mathbf{1}^\top c = 1$. The optimal transport problem can be written as, for $m = n^2$, and $A \in \{0,1\}^{2n \times m}, b \in \mathbb{R}^{2n}_{\geq 0}$, for $A$ the (unsigned) *edge-incidence matrix* of the underlying bipartite graph and $b$ the concatenation of $r$ and $c$.

$$\min_{x \in \Delta^m, Ax = b} d^\top x. \tag{2}$$

$$A = \begin{pmatrix} 1 & 1 & 1 & 0 & 0 & 0 & 0 & 0 & 0 \\ 0 & 0 & 0 & 1 & 1 & 1 & 0 & 0 & 0 \\ 0 & 0 & 0 & 0 & 0 & 0 & 1 & 1 & 1 \\ 1 & 0 & 0 & 1 & 0 & 0 & 1 & 0 & 0 \\ 0 & 1 & 0 & 0 & 1 & 0 & 0 & 1 & 0 \\ 0 & 0 & 1 & 0 & 0 & 1 & 0 & 0 & 1 \end{pmatrix}, \ b = \begin{pmatrix} 1/3 \\ 1/3 \\ 1/3 \\ 1/3 \\ 1/3 \\ 1/3 \end{pmatrix}.$$

Figure 1: Edge-incidence matrix $A$ of a $3 \times 3$ bipartite graph and uniform demands.

In particular, $A$ is the 0-1 matrix on $V \times E$ such that $A_{ve} = 1$ iff $v$ is an endpoint of edge $e$. We summarize some additional properties of the constraint matrix $A$ and vector $b$.

**Fact 2.1.** *A, b have the following properties.*

1. *$A \in \{0,1\}^{2n \times m}$ has 2-sparse columns and $n$-sparse rows. Thus $\|A\|_{1 \to 1} = 2$.*

2. *$b^\top = \begin{pmatrix} r^\top & c^\top \end{pmatrix}$, so that $\|b\|_1 = 2$.*

3. *$A$ has $2n^2$ nonzero entries.*

Section 4 recalls the proof of the following theorem, which first appeared in [AWR17].

**Theorem 2.2** (Rounding guarantee, Lemma 7 in [AWR17]). *There is an algorithm which takes $\tilde{x}$ with $\|A\tilde{x} - b\|_1 \leq \delta$ and produces $\hat{x}$ in $O(n^2)$ time, with*

$$A\hat{x} = b, \|\tilde{x} - \hat{x}\|_1 \leq 2\delta.$$

We now show how the rounding procedure gives a roadmap for our approach. Consider the following $\ell_1$ regression objective over the simplex (a similar penalized objective appeared in [She13]):

$$\min_{x \in \Delta^m} d^\top x + 2\|d\|_\infty \|Ax - b\|_1. \tag{3}$$

We show that the penalized objective value is still OPT, and furthermore any approximate minimizer yields an approximate transport plan.

**Lemma 2.3** (Penalized $\ell_1$ regression). *The value of (3) is OPT. Also, given $\tilde{x}$, an $\epsilon$-approximate minimizer to (3), we can find $\epsilon$-approximate transportation plan $\hat{x}$ in $O(n^2)$ time.*

*Proof.* Recall OPT $= \min_{x \in \Delta^m, Ax=b} d^\top x$. Let $\tilde{x}$ be the minimizing argument in (3). We claim there is some optimal $\tilde{x}$ with $A\tilde{x} = b$; clearly, the first claim is then true. Suppose otherwise, and let $\|A\tilde{x} - b\|_1 = \delta > 0$. Then, let $\hat{x}$ be the result of the algorithm in Theorem 2.2, applied to $\tilde{x}$, so that $A\hat{x} = b, \|\tilde{x} - \hat{x}\|_1 \leq 2\delta$. We then have

$$d^\top \hat{x} + 2\|d\|_\infty \|A\hat{x} - b\|_1 = d^\top(\hat{x} - \tilde{x}) + d^\top \tilde{x} \leq d^\top \tilde{x} + \|d\|_\infty \|\hat{x} - \tilde{x}\|_1 \leq d^\top \tilde{x} + 2\|d\|_\infty \delta.$$

The objective value of $\hat{x}$ is no more than of $\tilde{x}$, a contradiction. By this discussion, we can take any approximate minimizer to (3) and round it to a transport plan without increasing the objective. $\quad\square$

Section 3 proves Theorem 2.4, which says we can efficiently find an approximate minimizer to (3).

**Theorem 2.4** (Approximate $\ell_1$ regression over the simplex). *There is an algorithm (Algorithm 1) taking input $\epsilon$, which has $O((\|d\|_\infty \log n \log \gamma)/\epsilon)$ parallel depth for $\gamma = \log n \cdot \|d\|_\infty / \epsilon$, and total work $O(n^2(\|d\|_\infty \log n \log \gamma)/\epsilon)$, and obtains $\tilde{x}$ an $\epsilon$-additive approximation to the objective in (3).*

We will approach proving Theorem 2.4 through a primal-dual viewpoint, in light of the following (based on the definition of the $\ell_1$ norm):

$$\min_{x \in \Delta^m} d^\top x + 2\|d\|_\infty \|Ax - b\|_1 = \min_{x \in \Delta^m} \max_{y \in [-1,1]^{2n}} d^\top x + 2\|d\|_\infty (y^\top Ax - b^\top y). \tag{4}$$

Further, a low-*duality gap* pair to (4) yields an approximate minimizer to (3).

**Lemma 2.5** (Duality gap to error). *Suppose $x, y$ is feasible ($x \in \Delta^m, y \in [-1,1]^{2n}$), and for any feasible $u, v$,*

$$\left(d^\top x + 2\|d\|_\infty (v^\top Ax - b^\top v)\right) - \left(d^\top u + 2\|d\|_\infty (y^\top Au - b^\top y)\right) \leq \delta.$$

*Then, we have $d^\top x + 2\|d\|_\infty \|Ax - b\|_1 \leq \delta + \text{OPT}$.*

*Proof.* The result follows from maximizing over $v$, and noting that for the minimizing $u$,

$$d^\top u + 2\|d\|_\infty (y^\top Au - b^\top y) \leq d^\top u + 2\|d\|_\infty \|Au - b\|_1 = \text{OPT}.$$

$\quad\square$

Correspondingly, Section 3 gives an algorithm which obtains $(x, y)$ with bounded duality gap within the runtime of Theorem 2.4.

## 2.2 Notation

$\mathbb{R}_{\geq 0}$ is the nonnegative reals. $\mathbf{1}$ is the all-ones vector of appropriate dimension when clear. The probability simplex is $\Delta^d \overset{\text{def}}{=} \{v \mid v \in \mathbb{R}_{\geq 0}^d, \mathbf{1}^\top v = 1\}$. We say matrix $X$ is in the simplex of appropriate dimensions when its (nonnegative) entries sum to one.

$\|\cdot\|_1$ and $\|\cdot\|_\infty$ are the $\ell_1$ and $\ell_\infty$ norms, i.e. $\|v\|_1 = \sum_i |v_i|$ and $\|v\|_\infty = \max_i |v_i|$. When $A$ is a matrix, we let $\|A\|_{p \to q}$ be the matrix operator norm, i.e. $\sup_{\|v\|_p = 1} \|Av\|_q$, where $\|\cdot\|_p$ is the $\ell_p$ norm. In particular, $\|A\|_{1 \to 1}$ is the largest $\ell_1$ norm of a column of $A$.

Throughout $\log$ is the natural logarithm. For $x \in \Delta^d$, $h(x) = \sum_{i \in [d]} x_i \log x_i$ is (negative) entropy where $0 \log 0 = 0$ by convention. It is well-known that $\max_{x \in \Delta^d} h(x) - \min_{x \in \Delta^d} h(x) = \log d$.

We also use the Bregman divergence of a regularizer and the proximal operator of a divergence.

**Definition 2.6** (Bregman divergence). *For (differentiable) regularizer $r$ and $z, w$ in its domain, the* Bregman divergence *from $z$ to $w$ is*

$$V_z^r(w) \overset{\text{def}}{=} r(w) - r(z) - \langle \nabla r(z), w - z \rangle.$$

When $r$ is convex, the divergence is nonnegative and convex in the argument ($w$ in the definition).

**Definition 2.7** (Proximal operator). *For (differentiable) regularizer $r$, $z$ in its domain, and $g$ in the dual space (when the domain is in $\mathbb{R}^d$, so is the dual space), we define the* proximal operator *as*

$$\text{Prox}_z^r(g) \overset{\text{def}}{=} \text{argmin}_w \left\{ \langle g, w \rangle + V_z^r(w) \right\}.$$

Several variables have specialized meaning throughout. All graphs considered will be on $2n$ vertices with $m$ edges, i.e. $m = n^2$. $A \in \mathbb{R}^{2n \times m}$ is the edge-incidence matrix. $d$ is the vectorized cost matrix $C$. $b$ is the constraint vector, concatenating row and column constraints $r, c$. In algorithms for solving (4), $x$ and $y$ are primal (in a simplex) and dual (in a box) variables respectively. In Section 3, we adopt the linear programming perspective where the decision variable $x \in \Delta^m$ is a vector. In Section 4, for convenience we take the perspective where $X$ is an unflattened $n \times n$ matrix. $\mathcal{U}_{r,c}$ is the feasible polytope: when the domain is vectors, $\mathcal{U}_{r,c}$ is $x \mid Ax = b$, and when it is matrices, $\mathcal{U}_{r,c}$ is $X \mid X\mathbf{1} = r, X^\top \mathbf{1} = c$ (by flattening $X$ this is consistent).

## 3 Main Algorithm

This section describes our algorithm for finding a primal-dual pair $(x, y)$ with a small duality gap, with respect to the objective in (4), which we restate here for convenience:

$$\min_{x \in \mathcal{X}} \max_{y \in \mathcal{Y}} d^\top x + 2 \|d\|_\infty \left(y^\top A x - b^\top y\right), \quad \mathcal{X} \overset{\text{def}}{=} \Delta^m, \quad \mathcal{Y} \overset{\text{def}}{=} [-1, 1]^{2n}. \qquad \text{(Restatement of (4))}$$

Our algorithm is a specialization of the algorithm in [She17]. One of our technical contributions in this regard is an analysis of the algorithm which more closely relates it to the analysis of dual extrapolation [Nes07], an algorithm for finding approximate saddle points with a more standard analysis. In Section 3.1, we give the algorithmic framework and convergence analysis. In Section B.1, we provide analysis of an alternating minimization scheme for implementing steps of the procedure. The same procedure was used in [She17] which claimed without proof the linear convergence rate of the alternating minimization; we hope the analysis will make the method more broadly accessible to the optimization community. We defer many proofs to Appendix B.

### 3.1 Dual Extrapolation Framework

For an objective $F(x, y)$ convex in $x$ and concave in $y$, the standard way to measure the duality gap is to define the *gradient operator* $g(x, y) = (\nabla_x F(x, y), -\nabla_y F(x, y))$, and show that for $z = (x, y)$ and any $u$ on the product space, the *regret*, $\langle g(z), z - u \rangle$, is small. Correspondingly, we define

$$g(x, y) \overset{\text{def}}{=} \left(d + 2 \|d\|_\infty A^\top y, \ 2 \|d\|_\infty (b - Ax)\right).$$

The dual extrapolation framework [Nes07] requires a regularizer on the product space. The algorithm is simple to state; it takes two "mirror descent-like" steps each iteration, maintaining a state

$s_t$ in the dual space[6]. A typical setup is a Lipschitz gradient operator and a regularizer which is the sum of canonical strongly-convex regularizers in the norms corresponding to the product space $\mathcal{X}, \mathcal{Y}$. However, recent works have shown that this setup can be greatly relaxed and still obtain similar rates of convergence. In particular, [She17] introduced the following definition.

**Definition 3.1** (Area-convexity). *Regularizer $r$ is $\kappa$-area-convex with respect to operator $g$ if for any points $a, b, c$ in its domain,*

$$\kappa \left( r(a) + r(b) + r(c) - 3r\left(\frac{a+b+c}{3}\right)\right) \geq \langle g(b) - g(a), b - c \rangle. \tag{5}$$

Area-convexity is so named because $\langle g(b)-g(a), b-c \rangle$ can be viewed as measuring the "area" of the triangle with vertices $a, b, c$ with respect to some Jacobian matrix. In the case of bilinear objectives, the left hand side in the definition of area-convexity is invariant to permuting $a, b, c$, whereas the sign of the right hand side can be flipped by interchanging $a, c$, so area-convexity implies convexity. However, it does not even imply the regularizer $r$ is strongly-convex, a typical assumption for the convergence of mirror descent methods.

We state the algorithm for time horizon $T$; the only difference from [Nes07] is a factor of 2 in defining $s_{t+1}$, i.e. adding a $1/2\kappa$ multiple rather than $1/\kappa$. We find it of interest to explore whether this change is necessary or specific to the analysis of [She17].

---

**Algorithm 1** $\bar{w} = \texttt{Dual-Extrapolation}(\kappa, r, g, T)$: Dual extrapolation with area-convex $r$.

---
Initialize $s_0 = 0$, let $\bar{z}$ be the minimizer of $r$.
**for** $t < T$ **do**
    $z_t \leftarrow \text{Prox}_{\bar{z}}^r(s_t)$.
    $w_t \leftarrow \text{Prox}_{\bar{z}}^r\left(s_t + \frac{1}{\kappa}g(z_t)\right)$.
    $s_{t+1} \leftarrow s_t + \frac{1}{2\kappa}g(w_t)$.
    $t \leftarrow t + 1$.
**end for**
**return** $\bar{w} \stackrel{\text{def}}{=} \frac{1}{T}\sum_{t\in[T]} w_t$.

---

**Lemma 3.2** (Dual extrapolation convergence). *Suppose $r$ is $\kappa$-area-convex with respect to $g$. Further, suppose for some $u$, $\Theta \geq r(u) - r(\bar{z})$. Then, the output $\bar{w}$ to Algorithm 1 satisfies*

$$\langle g(\bar{w}), \bar{w} - u \rangle \leq \frac{2\kappa\Theta}{T}.$$

In fact, by more carefully analyzing the requirements of dual extrapolation we have the following.

**Corollary 3.3.** *Suppose in Algorithm 1, the proximal steps are implemented with $\epsilon'/4\kappa$ additive error. Then, the upper bound of the regret in Lemma 3.2 is $2\kappa\Theta/T + \epsilon'$.*

We now state a useful second-order characterization of area-convexity involving a relationship between the Jacobian of $g$ and the Hessian of $r$, which was proved in [She17].

**Theorem 3.4** (Second-order area-convexity, Theorem 1.6 in [She17]). *For bilinear minimax objectives, i.e. whose associated operator $g$ has Jacobian*

$$J = \begin{pmatrix} 0 & M^\top \\ -M & 0 \end{pmatrix},$$

*and for twice-differentiable $r$, if for all $z$ in the domain,*

$$\begin{pmatrix} \kappa\nabla^2 r(z) & -J \\ J & \kappa\nabla^2 r(z) \end{pmatrix} \succeq 0,$$

*then $r$ is $3\kappa$-area-convex with respect to $g$.*

Finally, we complete the outline of the algorithm by stating the specific regularizer we use, which first appeared in [She17]. We then prove its 3-area-convexity with respect to $g$ by using Theorem 3.4.

$$r(x, y) = 2 \|d\|_\infty \left( 10 \sum_{j \in [n]} x_j \log x_j + x^\top A^\top (y^2) \right), \tag{6}$$

where $(y^2)$ is entry-wise. To give some motivation for this regularizer, one $\ell_\infty$-strongly convex regularizer is $\frac{1}{2} \|y\|_2^2$, but over the $\ell_\infty$ ball, this regularizer has large range. The term $x^\top A^\top (y^2)$ in (6) captures the curvature required for strong-convexity locally, but has a smaller range due to the restrictions on $x, A$. The constants chosen were the smallest which satisfy the assumptions of the following Lemma 3.5.

**Lemma 3.5** (Area-convexity of the Sherman regularizer). *For the Jacobian $J$ associated with the objective in* (4) *and the regularizer $r$ defined in* (6)*, we have*

$$\begin{pmatrix} \nabla^2 r(z) & -J \\ J & \nabla^2 r(z) \end{pmatrix} \succeq 0.$$

We now give the proof of Theorem 2.4, requiring some claims in Appendix B.1 for the complexity of Algorithm 1. In particular, Appendix B.1 implies that although the minimizer to the proximal steps cannot be computed in closed form because of non-separability, a simple alternating scheme converges to an approximate-minimizer in near-constant time.

*Proof of Theorem 2.4.* The algorithm is Algorithm 1, using the regularizer $r$ in (6). Clearly, in the feasible region the range of the regularizer is at most $20 \|d\|_\infty \log n + 4 \|d\|_\infty$, where the former summand comes from the range of entropy and the latter $\|A^\top\|_\infty = 2$. We may choose $\Theta = O(\|d\|_\infty \log n)$ to be the range of $r$ to satisfy the assumptions of Lemma 3.2, since for all $u$, $\langle \nabla r(\bar{z}), \bar{z} - u \rangle \le 0 \Rightarrow V_{\bar{z}}^r(u) \le r(u) - r(\bar{z})$.

By Theorem 3.4 and Lemma 3.5, $r$ is 3-area-convex with respect to $g$. By Corollary 3.3, $T = 12\Theta/\epsilon$ iterations suffice, implementing each proximal step to $\epsilon/12$-additive accuracy. Finally, using Theorem B.5 to bound this implementation runtime concludes the proof. □

# 4 Rounding to $\mathcal{U}_{r,c}$

We state the rounding procedure in [AWR17] for completeness here, which takes a transport plan $\tilde{X}$ close to $\mathcal{U}_{r,c}$ and transforms it into a plan which exactly meets the constraints and is close to $\tilde{X}$ in $\ell_1$, and then prove its correctness in Appendix C. Throughout $r(X) \overset{\text{def}}{=} X\mathbf{1}, c(X) \overset{\text{def}}{=} X^\top \mathbf{1}$.

---

**Algorithm 2** $\hat{X} = \texttt{Rounding}(\tilde{X}, r, c)$: Rounding to feasible polytope

$X' \leftarrow \mathbf{diag}\left( \min\left( \frac{r}{r(\tilde{X})}, 1 \right) \right) \tilde{X}$.
$X'' \leftarrow X' \mathbf{diag}\left( \min\left( \frac{c}{c(X')}, 1 \right) \right)$.
$e_r \leftarrow r - \mathbf{1}^\top r(X''), e_c \leftarrow c - \mathbf{1}^\top c(X''), E \leftarrow \mathbf{1}^\top e_r$.
$\hat{X} \leftarrow X'' + \frac{1}{E} e_r e_c^\top$.
**return** $\hat{X}$.

---

# 5 Experiments

We show experiments illustrating the potential of our algorithm to be useful in practice, by considering its performance on computing optimal transport distances on the MNIST dataset and comparing against algorithms in the literature including APDAMD [LHJ19] and Sinkhorn iteration. All comparisons are based on the number of matrix-vector multiplications (rather than iterations, due to our algorithm's alternating subroutine), the main computational component of all algorithms considered.

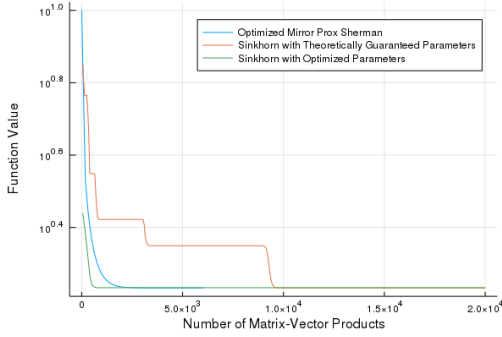
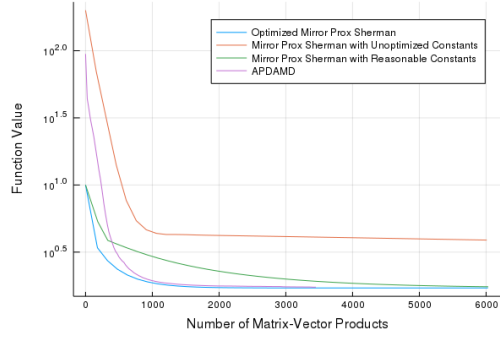

(a) Comparison with Sinkhorn iteration with different parameters.

(b) Comparison with APDAMD [LHJ19] with different parameters.

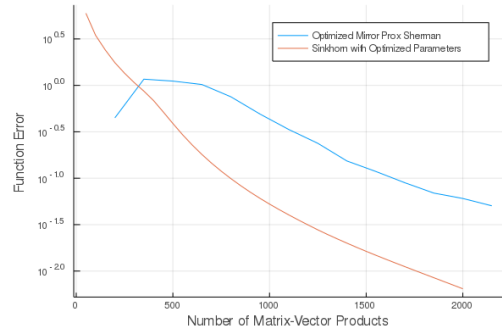
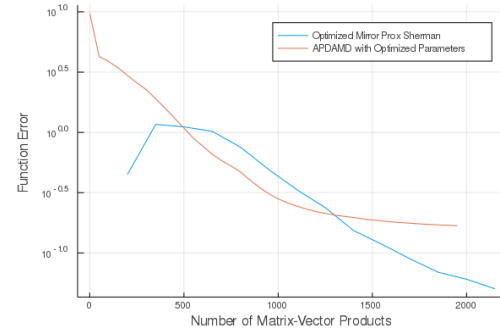

(a) Comparison with Sinkhorn iteration on 20 randomly chosen MNIST digit pairs.

(b) Comparison with APDAMD [LHJ19] on 20 randomly chosen MNIST digit pairs.

While our unoptimized algorithm performs poorly, slightly optimizing the size of the regularizer and step sizes used results in an algorithm with competitive performance to APDAMD, the first-order method with the best provable guarantees and observed practical performance. Sinkhorn iteration outperformed all first-order methods experimentally; however, an optimized version of our algorithm performed better than conservatively-regularized Sinkhorn iteration, and was more competitive with variants of Sinkhorn found in practice than other first-order methods.

As we discuss in our implementation details (Appendix D), we acknowledge that implementations of our algorithm illustrated are not the same as those with provable guarantees in our paper. However, we believe that our modifications are justifiable in theory, and consistent with those made in practice to existing algorithms. Further, we hope that studying the modifications we made (step size, using mirror prox [Nem04] for stability considerations), as well as the consideration of other numerical speedups such as greedy updates [AWR17] or kernel approximations [ABRW18], will become fruitful for understanding the potential of accelerated first-order methods in both the theory and practice of computational optimal transport.

**Acknowledgements**

We thank Jose Blanchet and Carson Kent for helpful conversations. AJ was supported by NSF Graduate Fellowship DGE-114747. AS was supported by NSF CAREER Award CCF-1844855. KT was supported by NSF Graduate Fellowship DGE-1656518.

## Footnotes

[1]Similarly to earlier works, we focus on square matrices; generalizations to rectangular matrices are straightforward.

[2]Our iterations consist of vector operations and matrix-vector products, which are easily parallelizable. Throughout $\|C\|_{\max}$ is the largest entry of $C$.

[3]We use "nearly linear" to describe complexities which have an $n^2\mathrm{polylog}(n)$ dependence on the dimension (where the size of input $C$ is $n^2$), and polynomial dependence on $\|C\|_{\max}, \epsilon^{-1}$.

[4]Because many of the objects defined in Section 2.2 are developed in Section 2.1, we postpone their statement, but refer the reader to Section 2.2 for any ambiguous definitions.

[5]We use $d$ because $C$ often arises from distances in a metric space, and to avoid overloading $c$.

[6]In this regard, it is more similar to the "dual averaging" or "lazy" mirror descent setup [Bub15].

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
