[Supplementary Material]

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

# A  Algorithm

We give the complete algorithm for approximating optimal transport distance to additive $\epsilon$ here. We assume $C \in \mathbb{R}_{\geq 0}^{n \times n}$ and $r, c \in \Delta^n$. Finally, we refer to blocks of variable $z$ on a product space as $z^x, z^y$, i.e. $z = (z^x, z^y)$. Again $r(X) \stackrel{\text{def}}{=} X\mathbf{1}$, $c(X) \stackrel{\text{def}}{=} X^\top \mathbf{1}$.

---

**Algorithm 3** $\hat{X} = \texttt{Optimal-Transport}(C, \epsilon, r, c)$: Produces $\epsilon$-approximate transportation plan

---

Vectorize $C$ to produce $d$.

Let $b$ be $r, c$ concatenated; let $A$ be the incidence matrix of a complete $n \times n$ bipartite graph.

$t \leftarrow 0$.

$x_0 \leftarrow \frac{1}{n^2}\mathbf{1}$, $y_0 \leftarrow \mathbf{0_{2n}}$.

$s_0^x \leftarrow \mathbf{0_{n^2}}$, $s_0^y \leftarrow \mathbf{0_{2n}}$.

$\Theta \leftarrow 20\,\|d\|_\infty \log n + 4\,\|d\|_\infty$.

**while** $d^\top x_{t+\frac{1}{2}} + 2\,\|d\|_\infty \left\| Ax_{t+\frac{1}{2}} - b \right\|_1 \leq -2\,\|d\|_\infty\, b^\top y_{t+\frac{1}{2}} + \max_j \left[ d + 2\,\|d\|_\infty\, A^\top y_{t+\frac{1}{2}} \right]_j + \epsilon$

**do**

  $t \leftarrow t+1$.

  $k \leftarrow 0$.

  $x_0' \leftarrow x_{t-\frac{1}{2}}$, $y_0' \leftarrow y_{t-\frac{1}{2}}$.

  **for** $0 \leq k < \left\lceil 24 \log \left( \left( \frac{88\|d\|_\infty}{\epsilon^2} + \frac{2}{\epsilon} \right) \Theta \right) \right\rceil$ **do**

    $x_k' \leftarrow \exp\left( \frac{1}{20\|d\|_\infty} s_t^x + \frac{1}{10} A^\top (y_{k-1}')^2 \right)$, $x_k' \leftarrow x_k' / \|x_k'\|_1$.

    $y_k' \leftarrow \min\left(1, \max\left(-1, \frac{-s_t^y}{4\|d\|_\infty Ax_k'}\right)\right)$. Operations are element-wise.

  **end for**

  $x_t \leftarrow x_k'$, $y_t \leftarrow y_k'$.

  $s_{t+\frac{1}{2}}^x \leftarrow s_t^x + \frac{1}{3}\left( d + 2\,\|d\|_\infty\, A^\top y_t \right)$.

  $s_{t+\frac{1}{2}}^y \leftarrow s_t^y + \frac{1}{3}\left( 2\,\|d\|_\infty\, (b - Ax_t) \right)$.

  $k \leftarrow 0$.

  $x_0' \leftarrow x_t$, $y_0' \leftarrow y_t$.

  **for** $0 \leq k < \left\lceil 24 \log \left( \left( \frac{88\|d\|_\infty}{\epsilon^2} + \frac{2}{\epsilon} \right) \Theta \right) \right\rceil$ **do**

    $x_k' \leftarrow \exp\left( \frac{1}{20\|d\|_\infty} s_{t+\frac{1}{2}}^x + \frac{1}{10} A^\top (y_{k-1}')^2 \right)$, $x_k' \leftarrow x_k' / \|x_k'\|_1$.

    $y_k' \leftarrow \min\left(1, \max\left(-1, \frac{-s_{t+\frac{1}{2}}^y}{4\|d\|_\infty Ax_k'}\right)\right)$. Operations are element-wise.

  **end for**

  $x_{t+\frac{1}{2}} \leftarrow x_k'$, $y_{t+\frac{1}{2}} \leftarrow y_k'$.

  $s_{t+1}^x \leftarrow s_t^x + \frac{1}{6}\left( d + 2\,\|d\|_\infty\, A^\top y_{t+\frac{1}{2}} \right)$.

  $s_{t+1}^y \leftarrow s_t^y + \frac{1}{6}\left( 2\,\|d\|_\infty\, (b - Ax_{t+\frac{1}{2}}) \right)$.

**end while**

Un-vectorize $x$ to produce $\tilde{X}$.

$X' \leftarrow \mathbf{diag}\left( \min\left( \frac{r}{r(\tilde{X})}, 1 \right) \right) \tilde{X}$.

$X'' \leftarrow X'\mathbf{diag}\left( \min\left( \frac{c}{c(X')}, 1 \right) \right)$.

$e_r \leftarrow r - \mathbf{1}^\top r(X'')$, $e_c \leftarrow c - \mathbf{1}^\top c(X'')$, $E \leftarrow \mathbf{1}^\top e_r$.

$\hat{X} \leftarrow X'' + \frac{1}{E} e_r e_c^\top$.

**return** $\hat{X}$.

---

We remark that there are a variety of termination conditions that can be useful in practice for the alternating minimization procedure. For example, a standard early-stopping condition based on the observed movement of consecutive iterates was very successful in practice (Appendix D).

# B Missing proofs from Section 3

In this section, we state missing proofs from Section 3. We provide the efficient implementation of the proximal steps required by Algorithm 1 in Appendix B.1.

*Proof of Lemma 3.2.* Our first step is to prove the following inequality:

$$\frac{1}{2\kappa}\langle g(w_t), w_t - \bar{z}\rangle \le \langle s_{t+1}, z_{t+1} - \bar{z}\rangle + V_{\bar{z}}^r(z_{t+1}) - \langle s_t, z_t - \bar{z}\rangle - V_{\bar{z}}^r(z_t). \tag{7}$$

Let $c_t = \frac{z_t + w_t + z_{t+1}}{3}$. The proof follows from minimality of $z_t$ with respect to $c_t$, minimality of $w_t$ with respect to $z_{t+1}$, and area-convexity (5) with respect to $z_t$, $w_t$, and $z_{t+1}$. Respectively,

$$\langle s_t, z_t\rangle + r(z_t) \le \langle s_t, c_t\rangle + r(c_t)$$

$$\langle s_t, w_t\rangle + \frac{1}{\kappa}\langle g(z_t), w_t\rangle + r(w_t) \le \langle s_t, z_{t+1}\rangle + \frac{1}{\kappa}\langle g(z_t), z_{t+1}\rangle + r(z_{t+1}) \tag{8}$$

$$\frac{1}{\kappa}\langle g(w_t) - g(z_t), w_t - z_{t+1}\rangle \le r(z_t) + r(w_t) + r(z_{t+1}) - 3r(c_t).$$

Adding three times the first equation to the third, rearranging, and using the definition of $c_t$, we have

$$\frac{1}{\kappa}\langle g(w_t) - g(z_t), w_t - z_{t+1}\rangle \le r(w_t) + r(z_{t+1}) - 2r(z_t) + \langle s_t, w_t + z_{t+1} - 2z_t\rangle.$$

Rearranging the second equation, we have

$$\frac{1}{\kappa}\langle g(z_t), w_t - z_{t+1}\rangle \le r(z_{t+1}) - r(w_t) + \langle s_t, z_{t+1} - w_t\rangle.$$

Adding these two equations, we have

$$\frac{1}{\kappa}\langle g(w_t), w_t - z_{t+1}\rangle \le 2r(z_{t+1}) - 2r(z_t) + \langle s_t, 2z_{t+1} - 2z_t\rangle.$$

Dividing by 2 and adding $\frac{1}{2\kappa}\langle g(w_t), z_{t+1} - \bar{z}\rangle$ to both sides, we obtain the desired (7). Now, define the potential function

$$\Phi_k = \frac{1}{2\kappa}\sum_{t=0}^{k-1}\langle g(w_t), w_t - \bar{z}\rangle - \langle s_k, z_k - \bar{z}\rangle - V_{\bar{z}}^r(z_k)$$

Then, by (7), $\Phi_k$ is nonincreasing in $k$. Therefore for any $u$, by the definition of $\Theta$,

$$\frac{1}{T}\sum_{t=0}^{T-1}\langle g(w_t), w_t - u\rangle \le \frac{1}{T}\sum_{t=0}^{T-1}\langle g(w_t), w_t - \bar{z}\rangle + \frac{1}{T}\sum_{t=0}^{T-1}\langle g(w_t), \bar{z} - u\rangle + \left(\frac{2\kappa\Theta}{T} - \frac{2\kappa V_{\bar{z}}(u)}{T}\right)$$

$$\le \frac{1}{T}\sum_{t=0}^{T-1}\langle g(w_t), w_t - \bar{z}\rangle + \frac{1}{T}\sum_{t=0}^{T-1}\langle g(w_t), \bar{z} - z_T\rangle + \left(\frac{2\kappa\Theta}{T} - \frac{2\kappa V_{\bar{z}}(z_T)}{T}\right)$$

$$= \frac{2\kappa}{T}\Phi_T + \frac{2\kappa\Theta}{T} \le \frac{2\kappa}{T}\Phi_0 + \frac{2\kappa\Theta}{T} = \frac{2\kappa\Theta}{T}.$$

The inequality on the second line used the definition of $z_T = \text{Prox}_{\bar{z}}^r\left(\frac{1}{2\kappa}\sum_{t\in[T-1]}g(w_t)\right)$, and the last inequality is $\Phi_T \le \Phi_0$. The conclusion follows from the definition of $g$ (because it is linear). □

*Proof of Corollary 3.3.* We see that (7) now holds up to $\frac{\epsilon'}{2\kappa}$ additive error, so that $\Phi_k$ is increasing by at most $\frac{\epsilon'}{2\kappa}$ each step. Thus, we obtain $\Phi_T \le \Phi_0 + \frac{T\epsilon'}{2\kappa}$, yielding the conclusion. □

*Proof of Lemma 3.5.* We scale both $r$ and $J$ down by $2\|d\|_\infty$, which does not affect positive-semidefiniteness. By computation we have (recalling all columns of $A$ have $\ell_1$ norm of 2)

$$\nabla^2 r(x, y) = \begin{pmatrix} 5\|A_{:j}\|_1\,\mathbf{diag}\left(\frac{1}{x_j}\right) & 2A^\top\mathbf{diag}\left(y_i\right) \\ 2\mathbf{diag}\left(y_i\right)A & 2\mathbf{diag}\left(A_i^\top x\right) \end{pmatrix}.$$

It suffices to show that for any vector $(a \quad b \quad c \quad d)$ we have

$$(a \quad b \quad c \quad d) \begin{pmatrix} 5\left\lVert A_{:j}\right\rVert_1 \mathbf{diag}\left(\frac{1}{x_j}\right) & 2A^\top \mathbf{diag}\left(y_i\right) & 0 & -A^\top \\ 2\mathbf{diag}\left(y_i\right)A & 2\mathbf{diag}\left(A_i^\top x\right) & A & 0 \\ 0 & A^\top & 5\left\lVert A_{:j}\right\rVert_1 \mathbf{diag}\left(\frac{1}{x_j}\right) & 2A^\top \mathbf{diag}\left(y_i\right) \\ -A & 0 & 2\mathbf{diag}\left(y_i\right)A & 2\mathbf{diag}\left(A_i^\top x\right) \end{pmatrix} \begin{pmatrix} a \\ b \\ c \\ d \end{pmatrix}$$

is nonnegative. Upon simplifying and gathering like terms, it suffices to show

$$\sum_{i,j} A_{ij} \left( \frac{5a_j^2}{x_j} + 4a_j b_i y_i + 2b_i^2 x_j - 2a_j d_i + 2c_j b_i + \frac{5c_j^2}{x_j} + 4c_j d_i y_i + 2d_i^2 x_j \right) \geq 0.$$

However, this is true for $y_i \in [-1, 1]$, since each coefficient groups into clearly nonnegative terms,

$$\left( \frac{4a_j^2}{x_j} + 4a_j b_i y_i + b_i^2 x_j \right) + \left( \frac{a_j^2}{x_j} - 2a_j d_i + d_i^2 x_j \right)$$

$$+ \left( \frac{4c_j^2}{x_j} + 4c_j d_i y_i + d_i^2 x_j \right) + \left( \frac{c_j^2}{x_j} + 2c_j b_i + b_i^2 x_j \right).$$

$\square$

## B.1 Alternating Minimization Analysis

In this section, we give the convergence analysis of an alternating minimization procedure for minimizing a function of the form (throughout this section, $r(x, y)$ is as in (6))

$$f(x, y) \stackrel{\text{def}}{=} \langle \xi, x \rangle + \langle \eta, y \rangle + r(x, y) \tag{9}$$

which is the type of minimization problem arising from steps of the form $\text{Prox}_z^\tau(g)$. As we will see, $f(x, y)$ is jointly convex. Throughout this section, let $x_{\text{OPT}}, y_{\text{OPT}}$ be the minimizer to $f$. Corollary 3.3 states that $O(\epsilon)$ additive error to $f$ gives the same asymptotic convergence rate in Algorithm 1. We will show that a simple alternating minimization scheme enjoys a linear rate of convergence in our setting; thus, roughly $O(\log \epsilon^{-1})$ iterations suffice. We first give a proof of a general condition which suffices for linear convergence.

**Lemma B.1.** *Suppose $f(x, y)$ is twice-differentiable and jointly convex, over the product space $\mathcal{X} \times \mathcal{Y}$. Consider the alternating minimization scheme,*

*1. $x_{k+1} \stackrel{\text{def}}{=} \text{argmin}_{x \in \mathcal{X}} f(x, y_k)$*

*2. $y_{k+1} \stackrel{\text{def}}{=} \text{argmin}_{y \in \mathcal{Y}} f(x_{k+1}, y)$*

*Further, suppose there are convex regions $\mathcal{X}_{k+1} \subseteq \mathcal{X}, \mathcal{Y}_k \subseteq \mathcal{Y}$ which contain $x_{k+1}, y_k$ respectively, such that for any $x' \in \mathcal{X}_{k+1}, y', y'' \in \mathcal{Y}_k$, and for some $\sigma \geq 1$,*

$$\nabla^2 f(x', y') \succeq \frac{1}{\sigma} \nabla_{yy}^2 f(x_{k+1}, y''), \tag{10}$$

*where $\nabla_{yy}^2$ is the Hessian with all but the yy block zeroed out. Then, for any $x^* \in \mathcal{X}_{k+1}, y^* \in \mathcal{Y}_k$,*

$$f(x_{k+1}, y_k) - f(x_{k+1}, y_{k+1}) \geq \frac{1}{\sigma} \left( f(x_{k+1}, y_k) - f(x^*, y^*) \right).$$

*Proof.* Let $\tilde{y} = \left(1 - \frac{1}{\sigma}\right) y_k + \frac{1}{\sigma} y^*$. We will prove instead that

$$f(x_{k+1}, y_k) - f(x_{k+1}, \tilde{y}) \geq \frac{1}{\sigma} \left( f(x_{k+1}, y_k) - f(x^*, y^*) \right),$$

from which the conclusion will follow since $f(x_{k+1}, y_{k+1}) \leq f(x_{k+1}, \tilde{y})$. Note by definition of $\tilde{y}$, as well as optimality of $x_{k+1}$ which implies $0 \geq \langle \nabla_x f(x_{k+1}, y_k), x_{k+1} - x^* \rangle$,

$$\langle \nabla_y f(x_{k+1}, y_k), y_k - \tilde{y} \rangle = \frac{1}{\sigma} \langle \nabla_y f(x_{k+1}, y_k), y_k - y^* \rangle \geq \frac{1}{\sigma} \langle \nabla f(x_{k+1}, y_k), z_{k+\frac{1}{2}} - z^* \rangle \tag{11}$$

where $z_{k+\frac{1}{2}} \stackrel{\text{def}}{=} (x_{k+1}, y_k)$ and $z^* \stackrel{\text{def}}{=} (x^*, y^*)$. Further, let $y_\alpha \stackrel{\text{def}}{=} (1-\alpha)y_k + \alpha y^*$, $\tilde{y}_\alpha \stackrel{\text{def}}{=} (1-\alpha)y_k + \alpha\tilde{y}$, and $x_\alpha \stackrel{\text{def}}{=} (1-\alpha)x_{k+1} + \alpha x^*$. Then, by Taylor expansion we have $f(x_{k+1}, y_k) - f(x_{k+1}, \tilde{y})$ equals

$$\langle \nabla_y f(x_{k+1}, y_k), y_k - \tilde{y} \rangle - \int_0^1 \int_0^\beta (\tilde{y} - y_k)^\top \nabla_{yy}^2 f(x_{k+1}, \tilde{y}_\alpha)(\tilde{y} - y_k) d\alpha d\beta$$

$$\geq \frac{1}{\sigma} \langle \nabla f(x_{k+1}, y_k), z_{k+\frac{1}{2}} - z^* \rangle - \frac{1}{\sigma^2} \int_0^1 \int_0^\beta (y^* - y_k)^\top \nabla_{yy}^2 f(x_{k+1}, \tilde{y}_\alpha)(y^* - y_k) d\alpha d\beta$$

$$\geq \frac{1}{\sigma} \left( \langle \nabla f(x_{k+1}, y_k), z_{k+\frac{1}{2}} - z^* \rangle - \int_0^1 \int_0^\beta (z^* - z_{k+\frac{1}{2}})^\top \nabla^2 f(x_\alpha, y_\alpha)(z^* - z_{k+\frac{1}{2}}) d\alpha d\beta \right)$$

$$= \frac{1}{\sigma} \left( f(x_{k+1}, y_k) - f(x^*, y^*) \right).$$

In the first inequality, we used (11) and the definition of $\tilde{y}$, and in the second we used (10) (since $x_\alpha \in \mathcal{X}_{k+1}, y_\alpha, \tilde{y}_\alpha \in \mathcal{Y}_k$ by convexity). $\qquad\square$

We now give a helper lemma specialized to the particular $f$ in (9), which will be used in the proof of convergence.

**Lemma B.2.** *For some $x_{k+1}, y_k$, let $\mathcal{X}_{k+1} = \left\{ x \mid x \geq \frac{1}{2}x_{k+1} \right\}$ where the inequality is entrywise, and let $\mathcal{Y}_k$ be the entire domain of $y$ (i.e. $\mathcal{Y}$). Then for any $x' \in \mathcal{X}_{k+1}, y', y'' \in \mathcal{Y}_k$,*

$$\nabla^2 r(x', y') \succeq \frac{1}{12} \nabla_{yy}^2 r(x_{k+1}, y'').$$

*Proof.* Recall that (since $\|A_{:j}\|_1 = 2$)

$$\nabla^2 r(x, y) = 2\|d\|_\infty \begin{pmatrix} 5\|A_{:j}\|_1 \operatorname{\mathbf{diag}}\left(\frac{1}{x_j}\right) & 2A^\top \operatorname{\mathbf{diag}}(y_i) \\ 2\operatorname{\mathbf{diag}}(y_i) A & 2\operatorname{\mathbf{diag}}(A_i^\top x) \end{pmatrix}.$$

Consider the diagonal approximation

$$D(x) = 2\|d\|_\infty \begin{pmatrix} \|A_{:j}\|_1 \operatorname{\mathbf{diag}}\left(\frac{1}{x_j}\right) & 0 \\ 0 & \operatorname{\mathbf{diag}}(A_i^\top x) \end{pmatrix}.$$

We claim for any $y$,

$$D(x) \preceq \nabla^2 r(x, y) \preceq 6D(x). \tag{12}$$

To see this, consider the quadratic forms with respect to some vector $(u \quad v)$:

$$(u \quad v) \nabla^2 r(x, y) \begin{pmatrix} u \\ v \end{pmatrix} = 2\|d\|_\infty \sum_{i,j} A_{ij} \left( \frac{5u_j^2}{x_j} + 4u_j v_i y_i + 2v_i^2 x_j \right),$$

$$(u \quad v) D(x) \begin{pmatrix} u \\ v \end{pmatrix} = 2\|d\|_\infty \sum_{i,j} A_{ij} \left( \frac{u_j^2}{x_j} + v_i^2 x_j \right).$$

Now (12) follows because for any $y_i \in [-1, 1]$, it's easy to verify

$$\frac{u_j^2}{x_j} + v_i^2 x_j \leq \frac{5u_j^2}{x_j} + 4u_j v_i y_i + 2v_i^2 x_j \leq 6 \left( \frac{u_j^2}{x_j} + v_i^2 x_j \right).$$

Therefore, to prove the lemma statement we can use

$$\nabla^2 r(x', y') \succeq D(x') \succeq \frac{1}{2} D(x_{k+1}) \succeq \frac{1}{12} \nabla_{yy}^2 r(x_{k+1}, y'').$$

The inequality $D(x') \succeq \frac{1}{2} D(x_{k+1})$ followed from the definition of $\mathcal{X}_{k+1}$, and the last inequality followed from $D(x_{k+1})$ spectrally dominating $\frac{1}{6} \nabla^2 r(x_{k+1}, y'')$, and restrictions of $D(x_{k+1})$ to the $yy$ block can only decrease the quadratic form. $\qquad\square$

We now give the proof of the linear rate of convergence.

**Lemma B.3.** *For $f(x, y)$ defined in* (9)*, the alternating minimization scheme*

1. $x_{k+1} \stackrel{\text{def}}{=} \text{argmin}_{x \in \mathcal{X}} f(x, y_k)$.

2. $y_{k+1} \stackrel{\text{def}}{=} \text{argmin}_{y \in \mathcal{Y}} f(x_{k+1}, y)$.

*decreases the function error $f(x_k, y_k) - f(x_{\text{OPT}}, y_{\text{OPT}})$ by a factor of at least $1/24$ in each iteration.*

*Proof.* We can apply Lemma B.1 with the sets defined in Lemma B.2, with $\sigma = 12$. On iteration $k$, consider picking the points $x^*, y^* = \frac{1}{2}(x_{k+1} + x_{\text{OPT}}), \frac{1}{2}(y_k + y_{\text{OPT}})$. Evidently, $x^* \in \mathcal{X}_{k+1}, y^* \in \mathcal{Y}_k$. Therefore, since $f(x_{k+1}, y_{k+1}) \geq f(x_{k+2}, y_{k+1})$,

$$f(x_{k+1}, y_k) - f(x_{k+2}, y_{k+1}) \geq f(x_{k+1}, y_k) - f(x_{k+1}, y_{k+1}) \geq \frac{1}{12}(f(x_{k+1}, y_k) - f(x^*, y^*)).$$

Furthermore, by convexity, we have

$$f(x_{k+1}, y_k) - f(x^*, y^*) \geq \frac{1}{2}(f(x_{k+1}, y_k) - f(x_{\text{OPT}}, y_{\text{OPT}})).$$

Finally, combining these two inequalities and rearranging,

$$\frac{23}{24}(f(x_{k+1}, y_k) - f(x_{\text{OPT}}, y_{\text{OPT}})) \geq f(x_{k+2}, y_{k+1}) - f(x_{\text{OPT}}, y_{\text{OPT}}).$$

Thus, by taking a $y$ step and then an $x$ step, we decrease the function error by a $1/24$ factor. □

Finally, we show that steps of the alternating minimization can be implemented in linear time.

**Lemma B.4.** *For $f(x, y)$ defined in* (9)*, we can implement the steps*

1. $x_{k+1} \stackrel{\text{def}}{=} \text{argmin}_x f(x, y_k)$.

2. $y_{k+1} \stackrel{\text{def}}{=} \text{argmin}_y f(x_{k+1}, y)$.

*restricted to the relevant domains, in time $O(n^2)$.*

*Proof.* Recall $A$ has $n^2$ nonzero entries, so a matrix-vector multiplication can be performed in this time. Computing $x$ in linear time is straightforward: it is defined by

$$\text{argmin}_x \langle \gamma, x \rangle + \sum_{j \in [n]} x_j \log x_j \text{ such that } x \in \Delta^m, \gamma \stackrel{\text{def}}{=} \frac{1}{20 \|d\|_\infty} \xi + \frac{1}{10} A^\top (y^2).$$

By examining the KKT conditions, it is clear that the minimizing $x$ is proportional to $\exp(-\gamma)$; computing $\gamma$ takes $O(n^2)$ time, as does the simplex projection. Similarly, computing $y$ in linear time is simple for fixed $x$: it is

$$\text{argmin}_y \langle \eta, y \rangle + \langle 2 \|d\|_\infty Ax, y^2 \rangle \text{ such that } y \in [-1, 1]^{2n},$$

which is coordinate-wise decomposable as minimizing a quadratic over an interval. □

**Theorem B.5** (Complexity of alternating minimization)**.** *We can obtain an $\epsilon/2$-approximate minimizer to the proximal steps required by Algorithm 1 to $\epsilon/2$ accuracy, with the regularizer of* (6) *and $\kappa = 3$, in $O(\log \gamma)$ parallelizable iterations for $\gamma = \log n \cdot \|d\|_\infty \cdot \epsilon^{-1}$, and $O(n^2 \log \gamma)$ total work.*

*Proof.* By Lemmas B.3 and B.4, we can spend $O(n^2)$ parallelizable work to decrease the suboptimality gap by a $1/24$ factor, so it remains to argue that the initial error is at most $\text{poly}(\log n, \|d\|_\infty, \epsilon^{-1})$ to show that implementing the proximal steps to additive error $\epsilon/2$ can be done in $O(\log \gamma)$ iterations. We show that this is true for implementing the proximal step for $z_t$; a

similar argument holds for $w_t$. To this end, note that by our setting of $\kappa$, for any $z$ where we let $g(z) = (g^x(z), g^y(z))$,

$$\frac{1}{2\kappa} \|g^x(z)\|_\infty = \frac{1}{6} \|d + 2\|d\|_\infty A^\top y\|_\infty \le \frac{\|d\|_\infty}{2},$$

$$\frac{1}{2\kappa} \|g^y(z)\|_1 = \frac{1}{6} \|2\|d\|_\infty (b - Ax)\|_1 \le \frac{4\|d\|_\infty}{3}.$$

Therefore, for $s_t = (s_t^x, s_t^y)$, by the triangle inequality, and $t \le 12\Theta/\epsilon$ the bound on the number of steps required where $\Theta$ is the range of $r$, we have

$$\|s_t^x\|_\infty \le t \cdot \frac{1}{2\kappa} \|g^x(z)\|_\infty \le \frac{6\|d\|_\infty \Theta}{\epsilon},$$

$$\|s_t^y\|_1 \le t \cdot \frac{1}{2\kappa} \|g^y(z)\|_1 \le \frac{16\|d\|_\infty \Theta}{\epsilon}.$$

A simple calculation yields $\Theta = 20\|d\|_\infty \log n + 4\|d\|_\infty$ upper bounds the range of $r$. Finally, let $x_t^*, y_t^*$ be the minimizer of the proximal objective,

$$\langle s_t^x, x \rangle + \langle s_t^y, y \rangle + r(x, y).$$

For any initialization $x_{\text{init}}, y_{\text{init}}$ to the alternating minimization, the suboptimality gap is given by

$$\langle s_t^x, x_{\text{init}} - x_t^* \rangle + \langle s_t^y, y_{\text{init}} - y_t^* \rangle + r(x_{\text{init}}, y_{\text{init}}) - r(x_t^*, y_t^*)$$

$$\le \|x_{\text{init}} - x_t^*\|_1 \|s_t^x\|_\infty + \|y_{\text{init}} - y_t^*\|_\infty \|s_t^y\|_1 + \Theta \le \left(\frac{44\|d\|_\infty}{\epsilon} + 1\right)\Theta.$$

Therefore, the total number of iterations required is bounded by $24\log\left(\left(\frac{88\|d\|_\infty}{\epsilon^2} + \frac{2}{\epsilon}\right)\Theta\right)$ as desired. $\qquad\square$

**Numerical precision.**   We also make a brief comment on bit-complexity issues which may arise when scaling exponentials. In particular, each of our alternating minimization steps of the form

$$\langle g, x \rangle + \sum_j x_j \log x_j \tag{13}$$

require exponentiating a potentially large vector $\log x - g$, and rescaling the vector to be on the simplex. The following lemma shows that we can implement this step with $O(\log n)$ bit-complexity, with a small polynomial (say $n^{-90}$) loss in the objective value. Because we may assume that $\epsilon^{-1}$ is bounded by say, $n^2$, else an interior point method achieves our stated runtime, the cumulative loss in objective value over all iterations due to limited precision is significantly less than $\epsilon$, and does not affect our asymptotic convergence rate. More precisely, we maintain $x$ implicitly through a vector $v$ which is $\log x$ up to a scaling, and show that by truncating $v$ to have its range of coordinates bounded by $O(\log n)$, the resulting simplex variable remains a high-precision minimizer to (13).

**Lemma B.6.** *Let $v \in \mathbb{R}^m$, and let $x \in \Delta^m$ be such that $x \propto \exp(v)$. Consider the following operation: let $j^* = \arg\min_j v_j$, and $j'$ be such that $v_{j'} < v_{j^*} - 100\log n$. Set $\hat{v} = v$ in every coordinate, except $\hat{v}_{j'} \leftarrow v_{j^*} - 100\log n$. Then, for $\hat{x} \propto \exp(\hat{v})$ in the simplex,*

$$\sum_j \hat{x}_j \log \hat{x}_j - \sum_j x_j \log x_j < n^{-95}, \langle g, \hat{x} - x \rangle < \|g\|_\infty n^{-95}.$$

*Proof.* Clearly, $\|\exp(v)\|_1 < \|\exp(\hat{v})\|_1$, since exp is monotone. Moreover,

$$\|\exp(\hat{v})\|_1 - \|\exp(v)\|_1 = \exp(v_{j^*} - 100\log n) - \exp(v_{j'}) < n^{-100}\exp(v_{j^*}) < n^{-100}\|\exp(v)\|_1.$$

Now, for every coordinate $j \ne j'$, this implies that $(1 - n^{-100})x_j < \hat{x}_j < x_j$. Thus,

$$\hat{x}_j \log \hat{x}_j - x_j \log x_j < -n^{-100}x_j \log x_j.$$

Furthermore, we have

$$\hat{x}_{j'} \log \hat{x}_{j'} - x_{j'} \log x_{j'} < -x_{j'} \log x_{j'} < 100n^{-100},$$

by $-x \log x$ is increasing for small $x$ and $x_{j'}$ is bounded by $n^{-100}$. Combining these estimates,

$$\sum_j \hat{x}_j \log \hat{x}_j - \sum_j x_j \log x_j < (100 + \log n)n^{-100} < n^{-95}$$

for $n > 10$. Similarly,

$$\langle g, \hat{x} - x \rangle \le \|g\|_\infty \left( \hat{x}_{j'} + \sum_{j \ne j'} n^{-100} x_j \right) \le \|g\|_\infty \, n^{-95}$$

for $n > 10$. $\qquad\square$

Thus, repeatedly applying Lemma B.6 every time we need to truncate a coordinate of $v$ due to finite bit precision, over the course of all iterations of the algorithm and all alternating minimization steps, the error incurred is negligible compared to the desired accuracy $\epsilon$ (where we also note $\|g\|_\infty$ for all $g$ we encounter is bounded by a small polynomial in $n$).

## C   Missing proofs from Section 4

In this section, we give the proof to Theorem 2.2.

**Theorem 2.2** (Rounding guarantee, Lemma 7 in [AWR17])**.** *There is an algorithm which takes $\tilde{x}$ with $\|A\tilde{x} - b\|_1 \le \delta$ and produces $\hat{x}$ in $O(n^2)$ time, with*

$$A\hat{x} = b, \|\tilde{x} - \hat{x}\|_1 \le 2\delta.$$

*Proof.* The algorithm is Algorithm 2. We adopt the alternative view of $\tilde{x}$ as a $n \times n$ matrix $\tilde{X}$ in the simplex, and define operations $r(X) = X\mathbf{1}, c(X) = X^\top \mathbf{1}$, recalling the first and last $n$ entries of $b$ are $r, c$, i.e. the row and column constraints. Recall we assume we have

$$\left\| r(\tilde{X}) - r \right\|_1 + \left\| c(\tilde{X}) - c \right\|_1 \le \delta.$$

Clearly all operations in Algorithm 2 take $O(n^2)$ time. To explain briefly, $X'$ is fixed so that its row sums are feasible (i.e. $X'\mathbf{1} \le r$) and $X''$ is fixed so that its column sums are feasible. Further, entrywise $X'' \le X' \le \tilde{X}$, so $X''$ is feasible. We first bound

$$d \overset{\text{def}}{=} \left\| X'' - \tilde{X} \right\|_1 = \left( \sum_{i:r_i(\tilde{X}) > r_i} r_i(\tilde{X}) - r_i \right) + \left( \sum_{j:c_j(X') > c_j} c_j(X') - c_j \right).$$

Note $\left\| r(\tilde{X}) - r \right\|_1 \ge \sum_{i:r_i(\tilde{X}) > r_i} r_i(\tilde{X}) - r_i$. Further, by $X' \le \tilde{X}$ entrywise,

$$\sum_{j:c_j(X') > c_j} c_j(X') - c_j \le \left\| c(\tilde{X}) - c \right\|_1.$$

Thus $d \le \delta$. $\hat{X} \in \mathcal{U}_{r,c}$, since $e_r, e_c \ge 0$ and $\mathbf{1}^\top e_r = \mathbf{1}^\top e_c = e$, so $\hat{X}\mathbf{1} = r, \ \hat{X}^\top \mathbf{1} = c$. Also,

$$\left\| \hat{X} - \tilde{X} \right\|_1 \le \left\| X'' - \tilde{X} \right\|_1 + \left\| \hat{X} - X'' \right\|_1 \le \delta + e.$$

Finally,

$$e = 1 - \mathbf{1}^\top X'' \mathbf{1} = 1 - \left( \mathbf{1}^\top \tilde{X}\mathbf{1} - d \right) = d.$$

Thus using $d \le \delta$ proves the claim. $\qquad\square$

# D   Experiment details

Here, we give the implementation details for the experimental results discussed in Section 5, and a brief justification of experimental decisions we made.

**Dataset.** For the first two figures in Section 5, we had the following experimental setup. We randomly sampled a pair of digits from the MNIST dataset corresponding to the digit 1, and added a small amount of background noise for numerical stability, as is standard in the literature [AWR17]. We downsampled the $28 \times 28$ pixel images to size $14 \times 14$ by skipping every other pixel to speed up experiments. Similar performances were observed across multiple random instances. Finally, the cost metric used was by Manhattan distance on the 2-dimensional grid.

For the second pair of figures, we randomly sampled 20 pairs of digits from the MNIST dataset where each pair corresponds to the same digit. As before we added a small amount of background noise for stability. As opposed to the previous comparison we ran all three algorithms on the true $28 \times 28$ pixel images. For each of the digit pairs we ran the Sinkhorn algorithm to high precision to obtain a baseline solution for comparison, and for each of the three algorithms tested we compared the value of the solutions obtained to this baseline. The plots compare the number of matrix-vector multiplies to the objective value error averaged over all 20 digit pairs. The metric is again the Manhattan distance over the 2-dimensional grid.

**Objective value.** For simplicity, in all cases we measured objective value by the overestimate presented in (4). By the proof of Lemma 2.3, this is an overestimate to the true objective after performing the rounding procedure in Algorithm 2. In practice, we observed that this overestimate was negligibly different from the objective after rounding.

**Sinkhorn implementation details.** We implemented the standard Sinkhorn algorithm, using different settings of $\eta^{-1}$. Sinkhorn iteration converges to an $\epsilon$-approximate transportation plan in theory when $\eta$ is very large, roughly $\log n/\epsilon$. However, in practice, it is observed that much smaller values of $\eta$ suffice for rapid convergence. We tracked the convergence of Sinkhorn iteration for $\eta = 70$ and $\eta = 5$, which we considered close to a theoretically guaranteed parameter and a much less conservative practical parameter, respectively. The optimized Sinkhorn algorithm converged at rates much faster than the predicted $\epsilon^{-2}$ rate on all experiments, outperforming all other methods, which we believe merits further investigation. Significantly larger values of $\eta$ led to numerical stability issues when computing $\exp(-\eta C)$.

**APDAMD implementation details.** We implemented the APDAMD algorithm (Algorithm 4 in [LHJ19]), with the quadratic regularizer (i.e. $\frac{1}{2\gamma} \|\lambda\|_2^2$). We observed that the amount of the quadratic regularizer added did not affect the practical convergence of the algorithm. A simple reason for this is because the algorithm builds in a more aggressive step-size strategy, because the pessimistic $\gamma = O(n)$ is often too conservative to be necessary in practice. The figure tracks APDAMD convergence with $\eta = 10^{-2}, \epsilon = 10^{-3}$.

**Mirror prox.** For numerical stability considerations, we implemented our algorithm as an instance of mirror prox [Nem04], another extragradient method which takes local iterations rather than accumulating a dual operator and taking steps with respect to some $\bar{z}$ (i.e. dual extrapolation). Although there is not a known proof of mirror prox convergence with an area-convex regularizer, we find this decision reasonable for several reasons. In general, variations of entropic mirror descent are well-known to be equivalent to their dual averaging versions; it is likely that a similar equivalence can be drawn between mirror prox and extragradient dual averaging, i.e. dual extrapolation. Furthermore, the standard proofs of dual extrapolation and mirror prox are quite similar; we believe it is likely that area-convexity results in convergence for mirror prox, although this merits further investigation.

**Termination.** We terminated our alternating minimization procedure when the movement of iterations in $\ell_1$ was negligible. Typically, we observed that 3-5 alternating steps sufficed for convergence.

**Step sizes.** We varied two parameters in our experiments: the step size $\frac{1}{\kappa}$ used in our extragradient algorithm, and the amount of entropy used in our regularizer (in the paper, we used 10 times entropy compared to the quadratic component $x^\top A^\top (y^2)$). One reason this may be reasonable in practice is similar to the observed behavior of the Sinkhorn iteration tuning the $\eta^{-1}$ parameter, and APDAMD performing a more-aggressive line search for the observed amount of regularizer necessary. We note that the need to tune the amount of entropy used in the regularizer is likely due to the analysis

not being tight in the constants, and with a tighter analysis, it may not be necessary to tune this parameter. To this end, we plotted the performance of three algorithm settings.

- In the "unoptimized constants", we set the constants to roughly those with theoretical guarantees, i.e. 10 times entropy and step size 1.

- In the "reasonably optimized constants", we set the amount of entropy to be 4, and the step size to be $\|d\|_\infty /3$, to offset the $\|d\|_\infty$ multiple of the regularizer used in our iterations. For smaller values of $\epsilon$, these settings compared favorably with APDAMD.

- In the "optimized constants", we set the amount of entropy at 3, and the step size at $\|d\|_\infty$. This setting outperformed APDAMD and was more competitive with Sinkhorn iteration.

**Discussion.** We believe multiple interesting avenues of exploration arise from our experiments.

- Sinkhorn with aggressively chosen $\eta$ outperformed all other methods we benchmarked against, and converged at rates faster than suggested by its known analyses. It may prove fruitful to study if further assumptions about practical instances explain this discrepancy.

- Directly accelerated methods such as APDAMD also exhibit $\epsilon^{-1}$ convergence rates, at the cost of a worse dependence on dimension. However, this worst-case dependence can be mitigated if the instance is favorable in practice, i.e. by choosing $\gamma \approx O(1)$. This was observed to be the case in our experiments for the MNIST dataset. It is interesting to see if a similar adaptive tuning applies to our method with provable guarantees.

- Our method did not exhibit instability when changing the amount of entropy in the regularizer, but it did exhibit vastly-improved convergence. It is possible that the amount of regularizer needed is not quite so large, perhaps through a more careful analysis.

- We did not benchmark against the greedy Sinkhorn method of [AWR17], or consider numerical speedups such as those in [ABRW18]. It remains open to explore if these practical speedups are applicable to first-order methods such as ours as well.