[Reviews · NeurIPS 2019]

Reviewer 1



Update after authors response: Thank you for the thorough responses to my questions. In light of these answers, I maintain that this paper should be accepted. ==== =1 I suggest revising the following claim in the abstract: “We… provide preliminary experimental evidence that our algorithm may enjoy improved practical performance” [L12-13] compared to the fastest practical algs, such as Sinkhorn. I suggest changing “improved practical performance” to “competitive practical performance”, as the experiments at best only appear to indicate this more modest claim, no? There is of course the issue of whether to optimize parameters when comparing algs—for both the proposed alg as well as Sinkhorn. This issue is brought up in the paper. However, it should be pointed out that Sinkhorn has only one parameter (entropy), whereas the proposed alg has two parameters (entropy and stepsize). Moreover, Sinkhorn’s parameter is very intuitive and in practice it is very easy to choose a reasonable value for it, so this parameter optimization is not a problem in practice for Sinkhorn. =2 The size of the experiments is extremely small, n=196. The authors write that they even downsampled from the original dataset which had n=784 (still very small) to speed up experiments. Sinkhorn easily scales to way larger sizes. So was this downsampling because the proposed alg does not scale well in n? Please include a plot demonstrating scalability in n, with n on the x-axis and runtime (for a fixed precision) on the y-axis. =3 Many runtimes reported in the previous work section are not scale invariant and thus seem very suspicious. E.g. ||C||_max^0.5/eps and ||C||_max/\eps^2 in L83 and Table 1. For scale invariance, runtimes for eps additive approx to OT should have the same dependence on ||C||_max as on 1/eps. Am I missing something? =4 Since there are exponentials in the proposed alg, please comment briefly on bit-complexity. In particular, is the bit-complexity only polylog in 1/eps, rather than poly in 1/eps? I ask since if the bit-complexity is e.g. 1/eps, then the n^2/eps runtime would really be n^2/eps^2, which is the same as Sinkhorn. =5 The authors state several times [e.g. L104,L275] that they believe their proposed alg can be sped up by using similar kernel-approx techniques as in [ABRW18]. How? The way this was done in [ABRW18] was low-rank approx of the kernel matrix exp(-eta*C), but that kernel matrix doesn’t seem to be relevant for the proposed alg. =6 I am confused why Appendix C exists. This is just re-proving a lemma from a previous paper, in the same way as originally done in that paper. Is there any difference? If not, this section should be deleted; just cite the original paper. =7 L227,L248: the inequalities seem to be in opposite directions? As a side comment, MNIST is perhaps not the best choice of dataset, since for OT between images, Sinkhorn iterations can be implemented exactly in \tilde{O}(n) time rather than n^2 using FFT. See e.g. Peyre-Cuturi’s book S4.3 Minor comments =L60: The use of entropic regularization for OT actually goes back half a century, see e.g. [Wilson, 1969, “The use of entropy maximising models…”] =L152: a minimizing argument =L153: there is some optimal x with Ax=b -> that all optimal x satisfy Ax=b =L187: Prox is missing r on top =L234: any -> all? =L251: n -> n^2 =References have many “??” =References: missing some capitalization, e.g. Newton, Lagrangian, Sinkhorn, Wasserstein, etc =App B: Lemmas B.1, B.2, B.3 are copy-and-pastes of Lemmas 3.2,3.3,3.5, respectively. I suggest removing the lemma statements in the appendix to shorten the paper. If not, at least re-title the lemmas to have the same numbering as main text (3.2,3.3,3.5) to avoid confusing the reader

Reviewer 2



--- After rebuttal I'm convinced with the correctness of Lemma 2.3. I also have a follow-up question: by taking (tilde x) as the optimal solution of Eq. (3), all inequalities become equal in L155. This implies that (tilde x) and (hat x) can only differ at where d attains maximum. Together with norm( (tilde x) - (hat x), 1) = delta and norm(A * (tilde x) - b,1) = 2*delta, it's possible to derive that (tilde x) = (hat x) under mild condition (e.g. max of d attains at d_{1n}=d_{n1}). That means Eq. (3) yields optimal solution that already satisfies the marginal constraint. This seems strong which made me think Lemma 2.3 is wrong. But now I can't seem to find a contradiction. Could you extend your analysis to address this? Some other corrections/suggestions: - Theorem 2.2 should have simplex constraint. - Page 4, L142: A has "2n^2" nonzero entries. - Pape 5, L152: min_{x \geq 0, Ax = b} - Page 5, L154: A(tilde x) - b (you've noticed this) - Appendix Lemma B.1 L415, better say you "add three times the first inequality to the third". - Lemma B.1 L422. Elaborate more on the last sentence. I believe it's because g is "affine" and the matrix is skew symmetric for you to make the LHS above equal to the LHS of the Lemma. - Lemma B.2. I think you need epsilon'/(2 kappa) additive error instead of epsilon' - Lemma B.3. The proof can be made simpler here. First r is area-convex, and hence convex, and hence its Hessian is nonnegative definite. Then the claim holds since you can split the matrix to [Hessian(r), 0; 0, Hessian(r)] which is nonnegative definite, and [0, -J;J, 0] which is skew symmetric. These two imply nonnegative definite. - Line 455: define z {k + 1/2} before using it, or use "where ...". Other thoughts: I like your reformulation of the original OT (1) into the min-max form (3). Although this is not the only way to solve (1) using primal-dual, it does yield an iteration complexity of n^2 (the 1/epsilon part is not that surprising since it's prima-dual, but you managed to show an n^2 which is not trivial). That said, I somehow feel that your attempt is a bit "too hard": you rely on the outer iteration complexity independent of n and inner iterations to converge linearly to get an overall n^2 complexity (so log epsilon inner complexity). In practice, I wonder if this worths the effort, especially when the contraction rate (upper bounded by 23/24 in your case) is a bit close to 1. That may give you a large constant in the complexity which may be even bigger than sqrt(n), where n couldn't be too large--an intrinsic issue with discretized OT. I raised my score and am inclined to accept, but hope you can address the issues I mentioned in your revised version. === Orignal review First of all, this paper is organized and presented in a very poor way. The main idea is merely solving the modified OT problem (3), which brings the marginal constraint in the original OT as a penalty term into the objective function. The modified problem is solved using a primal-dual algorithm, then the result is projected by an existing rounding algorithm to satisfy the marginal constraint exactly. This work is heavily based on Lemma 2.3, which states that an epsilon-solution of the modified problem is still an epsilon-solution of the original OT after the projection. However, this claim is problematic: the proof does not show Eq. (3) has the same optimal value OPT as the original OT--in fact, its optimal value should be smaller than OPT. In terms of algorithmic development, the paper just applies known dual extrapolation to solve the modified problem and the rounding algorithm for the projection, so the novelty is limited. The practical performance is even worse than some comparison methods. The presentation of the paper has a lot to improve. For example, Algorithm 1 should be introduced in companion with the considered problem with explanations on the terms. The regularization r in (6) should be presented before it's used. Notation section should be given before Overview where you actually use the notations extensively, etc.

Reviewer 3



For every mark (#), please refer to the comments of (5. improvements) * Clarity: The overall contribution of the paper is clear. The theoretical analysis and technical proofs are sound and easy to follow. The structure of Section 3 however could be improved (#1). * Quality / contribution The authors followed [BJKS18, Qua19] to write the Kantorovich OT problem as a positive LP problem which is shown to be equivalent to an l1 regression. The l1-linfty duality is then used to write the OT problem as a min-max problem. The dual extrapolation framework yields a bilinear saddle point problem. Adding a strongly convex regularizer (entropy + a term (#2)) allows to use the results of Sherman, 17 to derive the 1/eps convergence. Writing the OT problem as a positive LP and combining the work of [Sherman, 17] with entropy regularized OT is novel and interesting and could lead to promising subsequent results on regularized OT, besides the O(log(n)/ε) complexity (#3) However, (eventhough [Shu17] is carefully cited throughout the paper) since Shu17 already states the O(1/epsilon) convergence of a broader problem, the relative theoretical contribution of the paper should be made more explicit. Regardless of the obtained results, the empirical comparison is superficial and was not designed to reflect the theoretical findings (the linear convergence rate 1/epsilon vs 1/epsilon^2 for Sinkhorn for instance, nor the dependance on the dimension n). Moreover, the experiment consisted of only one computation of the Wasserstein distance on one single pair of downsampled MNIST images (14x14) which is not conclusive (#4).

[Author Response · NeurIPS 2019]

We thank the reviewers for their many helpful suggestions to improve the presentation. We first give general responses
to common issues raised. We hope our clarifications address concerns regarding the paper, and elevate your view of it.

1. Designing a parallel first-order $\tilde{O}(\epsilon^{-1})$ algorithm has been a well-studied open problem in computational OT and
our major contribution is its resolution. This problem has persisted despite extensive work on first-order methods and
varied reductions and though we lean on this literature, our specific objective $c^\top x + 2\|c\|_\infty \|Ax - b\|_1$ and method
used are new to OT and crucial to our improvement. A minor contribution is an analysis of area convexity closer to dual
extrapolation, and proving the complexity of a prox step, previously used without proof (3.7 [She17], which does not
follow from analysis in [Beck15]).

2. An objection to recent reduction-based algorithms with $\epsilon^{-1}$ rates was their impracticality; we would like to emphasize
our experiments' goal was to give preliminary evidence that our performance extends to real data. We hope this addresses
concerns that experiments did not show a practical outperformance of Sinkhorn, which we did not mean to claim.

3. Prior empirical work notes Sinkhorn outperforms its current worst-case bounds on real data [Cut13]. Common real
instances may have more structure bypassing the worst-case and allowing improved runtimes; we consider theoretical
guarantees with no additional assumptions. Regarding algorithms with better theoretical performance than Sinkhorn in
some regimes, we found our algorithm with tuned parameters outperformed APDAMD, the state of the art.

4. Using MNIST for experiments was for consistency with recent literature on computational OT. We acknowledge
additional structure of images yield further speedups, e.g. FFT, and modifications which do not improve theoretical
bounds may help in practice, e.g. coordinate methods ("Greenkhorn"), adaptivity. We focus on resolving the outstanding
issue of $\epsilon^{-1}$ dependence; we believe these optimizations merit interesting follow-up work, but it is outside our focus.

5. We thank the reviewers for their many suggestions regarding presentation order (introducing algorithm and regularizer
earlier, more motivation, necessity of App. C), agree they help readability, and will implement these in the next version.

**Reviewer 2** *Alg doesn't scale well?* Our implementation at submission time was designed only to compare iterations
and was inefficient. Since submission we improved the implementation and it now scales comparably with Sinkhorn at
least up to the full image dimensions $784 \times 784$. We will include experiments at this scale in the final version.

*Change "improved" to "competitive" in abstract.* We agree with this assessment and will implement this change.

*Two tunables?* In theory, the entropy constant is not an additional tunable, as the smallest satisfying B.3. It may not be
tight; we acknowledge the current analysis does not predict performance with a smaller constant. In practice, we believe
this should not be considered a second hyperparameter; we found a smaller constant, e.g. 3, sufficed for convergence.

*Previous runtime scale invariance.* We reported runtimes as claimed in prior work (e.g. Thms. 4.7, 4.9 [LHJ19]). We
investigated and believe the true dependence on $\|C\|_{\max}$ is as suggested by the reviewer, and will edit accordingly.

*Bit-complexity?* It is $O(\log(n))$ and we will add a discussion: assume $\epsilon = n^{-O(1)}$ (else IPM suffices), and maintain
variable $x$ proportional to $\exp(v)$ for $v$ with $O(\log(n))$ range without affecting correctness, by slightly modifying B.1.

*Kernel-approx [ABRW18]?* Our method is based on matrix-vector multiplies; thus we hope low-rank approximations to
the costs when more structure exists (kernel matrices arising from lower dimensions) can speed up iterations. We agree
the application is not immediately clear and will modify the language appropriately.

*L227,L248 inequalities wrong?* They are correct: $\Theta > r(u) - r(\bar{z})$ implies the divergence bound. We will clarify this.

**Reviewer 3** *Lemma 2.3 does not show (3) has the same optimal value?* To the best of our knowledge, 2.3 is correct
as stated. A similar penalized regression objective was used in the maximum flow literature, and the ideas behind
correctness are not new (2.2 [She13]). Such a penalized objective is new to OT which has typically considered an
$\epsilon$-regularized objective, whose value is not equal; our objective is not such a regularization. The proof: for any argument
to the modified objective, the value does not increase by rounding (to $A\tilde{x} = b$), so there exists optimal $\tilde{x}$ with $A\tilde{x} = b$.
OPT lower bounds the modified objective at $\tilde{x}$ by definition, and the minimizing argument of the OT objective achieves
it. We note $x$ in line 154 should be $\tilde{x}$, and we do not state that the OT plan achieves OPT; we will clarify these points.

**Reviewer 4** *Comparisons on more pictures? Average several trials?* We agree with the suggestion. We experimented
on other digits: across multiple trials we observed similar performance, and will include additional experiments.

*Is the full objective $\ell_1$-reg + regularizer? Clarify constants in regularization, motivate regularizer.* The objective we
optimize is only the $\ell_1$-regularized portion; the regularizer is used to define algorithm steps. Re: constants: $\kappa$ is the
largest which satisfies area convexity (3), and the entropy constant (10) is the smallest which makes the quadratic form
PSD in B.3. The regularizer $x^\top A(y^2)$ captures a "local version" of the smallest-width $\ell_\infty$-strongly-convex regularizer,
a quadratic (A.1 of [ST18]). Since we want to decrease regularizer size and $x \in \Delta^m$, this is a small regularizer which
captures enough local behavior of a quadratic allowing for area-convexity. We will add this discussion.

[Meta-Review · NeurIPS 2019]

The paper is accepted following a long discussion post rebuttal. Reviewers believe there is value in this paper, since it provides an improved result over the currently known bounds of the Sinkhorn algorithm using a different strategy. I do however share the concern expressed by most reviewers that the experimental section is underwhelming. The paper would greatly benefit from taking more seriously this aspect into account. In the setup considered by the authors the Sinkhorn algorithm can easily scale to 10k x 10k grids using the convolution trick, or in 3D in 200 x 200 x 200. The scales presented here are too small to be convincing. We understand this is not the main ingredient of the paper, but if the paper is to be presented at Neurips a diligent experimental evaluation is a big plus for the community.